# Large Language Models based Graph Convolution for Text-Attributed Networks

## Abstract

Text-attributed graph (TAG) tasks involve analyzing both structural information and textual attributes. Existing methods employ text embeddings as node features, and leverage structural information by employing Graph Neural Networks (GNNs) to aggregate features from neighbors. These approaches demand substantial computational resources and rely on two cascaded stages, resulting in a sub-optimal learning process and making them vulnerable to the influence of irrelevant neighboring nodes. The advancement of language models (LMs) presents new avenues for tackling this task without GNNs, leveraging their ability to process text attributes of both the target node and its important neighbors. Instead of using graph convolution modules, LMs can assign weights to these tokens based on relevance, enabling token-level weighted summarization. However, it is non-trivial to directly employ LMs for TAG tasks because assessing the importance of neighbor nodes involves both semantic and structural considerations. Additionally, the large search space presents efficiency issues for computing importance scores in a scalable manner. To this end, we propose a novel **S**emantic **K**nowledg**e** and **S**tructural Enri**ch**ment framework, namely **SKETCH**, to adapt LMs for TAG tasks by retrieving both structural and text-related content. Specifically, we propose a retrieval model that identifies neighboring nodes exhibiting similarity to the target node across two dimensions: structural similarity and text similarity. To enable efficient retrieval, we introduce a hash-based common neighbor estimation algorithm for structural similarity and a nearest-neighbor recalling algorithm for embedding similarity. These two similarity measures are then aggregated using a weighted rank aggregation mechanism. The text attributes of both the retrieved nodes and the target node provide effective descriptions of the target node and are used as input for the LM predictor. Extensive experiments demonstrate that SKETCH can outperform other baselines on three datasets with fewer resources.

## 1 Introduction

Text-attributed graphs (TAGs) are frequently encountered in various real-world scenarios, including academic networks, e-commerce platforms, and social networks (Tang et al., 2008; He & McAuley, 2016; Jin et al., 2023). The model is required to make the inference and prediction using the textual information contained in nodes and the graphical structures formed by the edges. Traditional pipelines use NLP techniques like bag-of-words and pre-trained models to embed text features and apply Graph Neural Networks (GNNs) (Wu et al., 2019; Veličković et al., 2018; Huang et al., 2022) for graph propagation. Recent studies leverage fine-tuning to learn more meaningful embeddings for downstream tasks and utilize the strong comprehension abilities of large language models. However, this cascaded framework presents a problem, as the text representations and graph structure are trained independently from their respective aspects, potentially resulting in sub-optimal integration between the two modalities (Duan et al., 2023). As a result, GNNs may not fully leverage the rich semantic contexts represented in the textual embeddings, and conversely, the text features may not adequately account for the structural nuances present in the graph (Zhou et al., 2020). This disconnect leads to inefficiencies and potentially hinders the performance of downstream TAG tasks that rely on both modalities, as the learning dynamics of the text and graph are not aligned. Consequently, the separate processing stages do not take into account the simultaneous optimization of the two data types, resulting in information loss and reduced robustness.

Besides, GNNs primarily depend on node-level aggregation via graph convolutions to compute weighted sums of neighboring features. While effective, this method may miss the rich semantic nuances in textual data. To address this limitation, we propose purely leveraging advanced language models to allow for weighted-sum computations not only at the node level but also at the token level, enhancing the representation learning of textual attributes. This shift enables a more granular understanding of the relationships between tokens, leading to improved flexibility and adaptability. The emergence of long context models offers the opportunity to convert graph structures into relation descriptions for long-text processing. Preserving long-range dependencies in text facilitates graph-based reasoning. By integrating extra text bodies with graph-analyzing techniques, we can supplement text relationships and their structural connections, enhancing predictions and insights from the graph. Purely using language models not only enhances aggregation flexibility but also improves textual information mining compared to cascaded models, as this approach allows the optimizer to learn attention weights for each token specifically. However, it's non-trivial to apply straightforwardly for two reasons: (*i*) Despite the increased token length, the vast number of nodes and edges still surpasses the text limit, making it impractical to provide the model with full graphical information. (*ii*) Text models can only process textual information and cannot replicate the graph-level search along the relation path. LLMs are good at understanding the text but may struggle with the relational context present in graphs. Therefore, identifying the underlying graph topology and filtering out relevant contexts through linked relationships is essential to this problem.

To this end, we introduce a novel graph-retrieval learning framework called SKETCH, which consists of two core modules: the semantic retrieval module and the structural retrieval module. Drawing on the principles that Graph Neural Networks are designed to capture not only the information from neighboring nodes but also features from distant nodes through a multi-layer propagation mechanism, SKETCH selects relevant semantic and graph-related contexts. It combines them into a long-context language model for predictions, providing rich contexts and enhancing understanding, which improves the model's capability to generate relevant responses and make a better inference.

**Contributions:**

- We introduce retrieval-enhanced learning for text-attributed graphs using long-context language models, allowing flexible token-level aggregation without relying on graph neural networks and shifting focus from traditional node-level aggregation.
- We propose the SKETCH framework, which improves learning by selectively integrating informative corpora from semantic and graph perspectives to extract richer information.
- To assess the structural relatedness between nodes during graph retrieval, we propose a standard on the number of common neighbors. To alleviate the significant computational burden, we introduce a novel hash-based method to approximate the extent of similarity.
- Extensive experiments show that our model excels in TAG learning, with SKETCH outperforming all state-of-the-art methods while requiring fewer computational resources. The studies assess the effectiveness of each module and the effects of various retrieval strategies.

## 2 APPROACH: SKETCH

**Notations.** Text-attributed graphs consider both text attributes and graph structure, unlike traditional text prediction and graph prediction tasks. A text-attributed graph is defined as $\mathcal{G} = (\mathcal{V}, \mathcal{E}, \mathcal{S})$, where $\mathcal{S}$ represents text attributes for each node. $\mathcal{V}$ denotes the set of nodes, $\mathcal{E}$ denotes edges between text nodes and $\mathcal{N}^k(v)$ denotes the $k$-hop neighbors of node $v$. Ground truth labels for a given text-attributed graph are denoted as $\boldsymbol{Y} = \{\boldsymbol{y}_1, \cdots, \boldsymbol{y}_{|S|}\}$, where $|S|$ is the size of the text-attributed nodes.

Our primary objective is to examine each anchor node within the graph to effectively identify the content that is most relevant and beneficial in enhancing the understanding of the associated text. As mentioned in the introduction, the complexity and richness of information at both the node and relationship levels require a nuanced analytical approach. Textual attributes offer valuable semantic insights into the meaning and context of the nodes, while structural relationships demonstrate how these nodes interact within the graph's topology. Therefore, our method leverages the inherent structure of the graph, treating all contained texts as valuable resources. In the following sections, we will detail our methodology for retrieving both semantically and structurally related corpora,

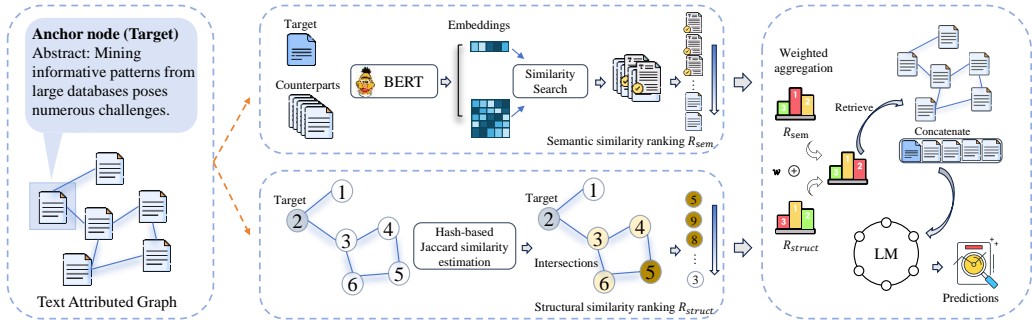

Figure 1: The SKETCH method for text-attributed graphs comprises two components: the Semantic Retrieval Module, which uses embedding similarity, and the Structural Retrieval Module, which employs Jaccard similarity. A weighted-rank aggregation mechanism combines their outputs, ranking the text of nodes, which are then fed into a language model for training and predictions.

enhancing our understanding of each anchor node and its context within the entire network. Here, we present a detailed illustration of the overall framework and its various components in Figure 1.

## 2.1 SEMANTIC RELATED RETRIEVAL

The structure of text-attributed graphs encompasses textual information from various nodes. Inspired by the concepts from Retrieval-Augmented Generation (RAG), we propose integrating additional corpus during the training process. These supplementary texts can significantly enhance the model's ability to make accurate predictions by providing essential context and knowledge. While some nodes are directly connected through edges, there are also nodes that, despite not being connected, may contain relevant information about the target node, referred to as the anchor node. Consider a research paper titled "Transfer Learning for Small Datasets in Medical Imaging." This paper addresses a specialized topic and is published in a niche journal, resulting in a limited number of direct citations. In this case, the text attributes of the paper including its abstract, keywords, and methodology—contain critical insights about "transfer learning" and "medical imaging." For instance, the methodologies proposed in the non-connected papers may introduce novel algorithms or frameworks that could enhance the explanation of the proposed technique.

To leverage this potential, we employ a global embedding similarity technique to retrieve useful nodes. This approach allows us to identify and extract information from both directly linked and indirectly related nodes, enhancing the overall relevance and comprehensiveness of the information associated with the anchor node. In our retrieval process, we begin by using a sentence-transformer to embed each piece of textual information into vector representations, as shown in Figure 2. These embeddings are then stored efficiently, allowing for quick access. To identify the most relevant content, we leverage the FAISS engine, which enables high-speed searching based on cosine similarity. Notably, even with a dataset containing hundreds of thousands of points, we can obtain results in just a few minutes. This efficiency ensures that we can quickly retrieve the most closely matching texts, streamlining the integration of relevant information into our system.

## 2.2 STRUCTURAL RELATED RETRIEVAL

### 2.2.1 DEFINING STRUCTURAL RELATEDNESS

This section focuses on retrieving structurally related nodes. However, each anchor node has numerous $k$-hop neighbors, making it impractical to include all in our analysis. Thus, we need to rank the importance of neighbors and select the most relevant ones. Previous research suggests that in graph learning, nodes with many common neighbors are often more closely related for several reasons. **Structural Similarity:** Nodes with many shared neighbors tend to be structurally similar, indicating similar roles or functions, particularly in social or biological networks. **Transitive Relationships:** Transitivity implies that if node A is connected to B and B is connected to C, A and C may also be related. Common neighbors signify potential transitive relationships, suggesting that nodes are indeed related. **Shared Context:** Common neighbors indicate that nodes share sim-

ilar contexts or environments. For instance, in a social network, two individuals with many mutual friends may have shared interests or activities, reflecting a closer relationship.

To better illustrate this pattern, we include a real-life example from our citation network dataset in Figure 3. The anchor node is labeled 'database', focusing on Kalman Filters, which may lead to its misclassification as a 'Machine Learning' algorithm. Here, additional context from cited papers is crucial. The neighboring node content, such as "applying adaptive filters for query processing in a distributed stream" and "techniques to query large data repositories efficiently", suggests that Kalman Filters are used to handle data streams and applied in the database field. In contrast, another connected node describing "Sensor networks have recently found many popular applications" serves a less relevant role. Analyzing their topological differences reveals that nodes with more common neighbors typically offer more relevant explanations. This aligns with the intuition that birds of a feather flock together and supports our previous experience that while many papers may be cited, some provide essential insights while others serve merely as supplementary information.

Based on this finding, we propose defining relatedness through intersection features using the Jaccard similarity coefficient, expressed as $J(A, B) = \frac{|N(A) \cap N(B)|}{|N(A) \cup N(B)|}$. In this formula, $J(A, B)$ represents the Jaccard similarity between nodes $A$ and $B$, where $N(A)$ denotes the set of neighbors of node $A$ and $N(B)$ the set of neighbors of node $B$. The numerator, $|N(A) \cap N(B)|$, indicates the number of common neighbors shared by the two nodes, while the denominator, $|N(A) \cup N(B)|$, represents the total number of unique neighbors across both nodes. By employing the Jaccard similarity, we can effectively capture the connectivity patterns that reflect structural relationships between nodes. This measure provides valuable insights critical for downstream graph-related tasks within our retrieval processes, facilitating a deeper understanding of the underlying graph structure.

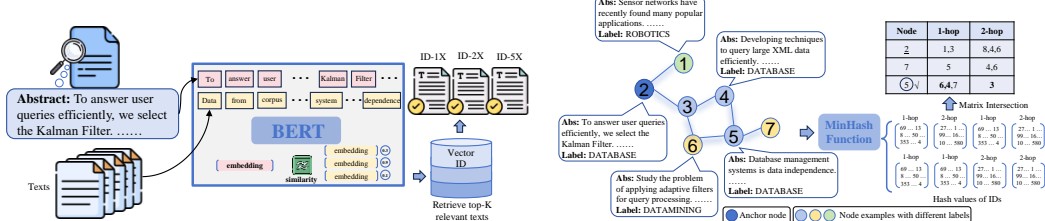

Figure 2: Overview of the semantic retrieval process. This figure illustrates in detail how we conduct a global search for similar nodes based on their embeddings, leveraging FAISS (Facebook AI Similarity Search) to efficiently identify and select the most similar items from the entire network.

Figure 3: Process of transforming neighborhood IDs into hash values for enhanced structural retrieval. The figure illustrates that the number of common neighbors is a crucial indicator for determining node relatedness among the graph and can be efficiently computed using matrix operations."

### 2.2.2 HASH-BASED JACCARD SIMILARITY ESTIMATION

As discussed in previous sections, the importance ranking of $k$-hop neighbors is determined by the proportion of common neighbors; more common neighbors indicate a closer relationship. This part formally presents our method for estimating $k$-hop Jaccard similarities for each pair of nodes. The typical approach involves recording each node's neighbors and incrementally counting the Jaccard. However, this method is time-consuming due to the varying number of neighbors per node and the large overall number of nodes, resulting in extensive iterations. Calculating multi-hop Jaccard, e.g. the Jaccard between the one-hop neighbors of an anchor node and the two-hop neighbors of other nodes, will further complicate the computation. Monte Carlo simulation is a possible solution, but it remains inefficient as it requires large samples for even a rough estimate.

**Similarity Estimation Using Minhash Functions.** To properly address the inefficiency associated with direct similarity computation, we plan to map neighborhood IDs into dense sketches and estimate the Jaccard extent in a lower-dimensional space. We choose Minhash Charikar (2002) functions as the mapping functions based on their properties. In this technique, we formulate the $k$-hop neighbors of a node $v_q \in \mathcal{V}$ as a set $\mathcal{N}(v_q)$. Next, we randomly hash every $v \in \mathcal{N}(v_q)$ in the set to an integer $h(v) \in [B]$. Here $h$ is a universal hashing function Carter & Wegman (1977). Next,

we take the minimum value of $h(v)$ for all $v \in \mathcal{N}(v_q)$ as the hash signature of the $k$-hop neighbors $\mathcal{N}(v_q)$. Next, we show that this Minhash function serves as an unbiased estimator of the Jaccard between the $k$-hop neighbors of two nodes.

**Definition 2.1** (Minhash for $k$-hop Neighbors Jaccard Estimation). *Let $\mathcal{V}$ denote the nodes in a graph $\mathcal{G} = (\mathcal{V}, \mathcal{E}, \mathcal{S})$. Let $\mathcal{N}^k(v)$ denote a set of the $k$-hop neighbors of node $v \in \mathcal{V}$. Let $h : \mathcal{V} \to [B]$ denote a universal hashing function that maps a node $v \in \mathcal{V}$ to an integer in range $[B]$. We define a Minhash function* Minhash *on $\mathcal{N}^k(v)$ as:*

$$\mathsf{Minhash}(\mathcal{N}^k(v)) = \min h(\mathcal{N}^k(v)).$$

*Moreover, based on the propriety of Minhash function Charikar (2002), we see that for $v_1, v_2 \in \mathcal{V}$*

$$\Pr[\mathsf{Minhash}(\mathcal{N}^k(v_1)) = \mathsf{Minhash}(\mathcal{N}^k(v_2))] = \frac{|\mathcal{N}^k(v_1) \cap \mathcal{N}^k(v_2)|}{|\mathcal{N}^k(v_1) \cup \mathcal{N}^k(v_2)|} = \mathcal{J}(\mathcal{N}^k(v_1), \mathcal{N}^k(v_2)).$$

As shown in the definition, Minhash is a locality-sensitive hashing function (Indyk & Motwani, 1998; Datar et al., 2004; Andoni et al., 2014; Andoni & Razenshteyn, 2015; Andoni et al., 2017). The collision probability of Minhash is equal to the Jaccard similarity of two $k$-hop neighbor sets. As a result, we use the collision of two Minhash signatures as an unbiased estimator to the Jaccard similarity between two $k$-hop neighbor sets $\mathcal{N}^k(v_1), \mathcal{N}^k(v_2)$.

**Multi-hop Similarity Estimation.** Since we are required to estimate Jaccard across $k$ hops instead of just one-hop neighbors, we extend the aforementioned method into $k$ dimensions. Our algorithm (see Algorithm 1) begins by extracting $\mathcal{N}^k(v)$, the sets of $k$-hop neighbors of the anchor node $v$. Next, we apply $R$ independent MinHash functions to generate $R$ hash values for every $s_{v,k} \in \mathcal{N}^k(v)$, repeating this process $l$ times. This effectively transforms the neighbor ID sequences into a hash sequence of length $l$, which we denote as $H$. To simplify, we compute 2-hop intersections by examining the one-hop and two-hop neighbors of each node, resulting in four combinations: one-hop with one-hop, one-hop with two-hop, two-hop with one-hop, and two-hop with two-hop neighbors. This approach can easily be extended to higher-hop manipulations.

For each node, we repeat the hash sequences corresponding to each hop. For example, let $h_{1,1}$, $h_{1,2}$ represent the one-hop and two-hop neighbors of the first node, respectively. The concatenated vector for node 1 would be $[h_{1,1}, h_{1,1}, h_{1,2}, h_{1,2}]$. We then arrange the hash vectors of all nodes according to the pattern $[[h_{1,1}, h_{1,2}, h_{1,1}, h_{1,2}], ..., [h_{k,1}, h_{k,2}, h_{k,1}, h_{k,2}]]$. By calculating the number of equal hash values present in each row, we effectively capture the cross-combinations that occur during multi-hop intersections. This process can be executed rapidly by leveraging the broadcasting capabilities of PyTorch in matrix operations. In contrast, common methods often require one-by-one iteration due to the irregularity in the number of neighbors, which significantly slows down the speed of computation. For simplicity, we previously set the hash sequence length for each vector to $l$, indicating equal importance among all four possible intersections. To control some weights over specific intersections, particularly the crucial one-hop intersections, we can shorten some other vectors to a length $m$, where $m < l$. This adjustment decreases the frequency of "collision" for the corresponding $k$-th hop intersection. Lastly, we rank the importance of neighbors for each anchor node by its row-wise sum; a higher value indicates a greater likelihood of sharing more neighbors.

## 2.3 AGGREGATED LEARNING OF RETRIEVED CONTENT

The combination of the two aforementioned modules aims to acquire relevant information from different dimensions, thereby enhancing the training effectiveness of the long-context model. Semantic retrieval focuses on identifying content that is semantically similar to the anchor node text, ensuring that we capture deep connections between texts. In contrast, structure retrieval emphasizes the links between texts, paying attention to the flow of information. After obtaining the relevant texts, we rank and filter them to select the most significant ones, which are then connected to the anchor node text to create a rich context for the long-context model. Specifically, we propose a weighted method to combine two different similarity metrics: semantic similarity ranking and structural similarity rankings. To create a unified scoring system, we introduce a hyperparameter $w$, which allows for the adjustment of weights of the overall score. The composite score is calculated using the formula $G = R_{\text{sem}} + w \cdot R_{\text{struct}}$, where $R_{\text{sem}}$ is the semantic rank and $R_{\text{struct}}$ is the structural similarity ranking. This approach allows for greater flexibility and enhances the capability of our evaluation method. It

is similar to the Retrieval-Augmented Generation pipeline, as our goal is to supplement information to improve model training. Our retrieval strategy draws inspiration from the fundamental principles of graph neural networks, where the core concept revolves around propagating information across edges to aggregate insights from spatially close and far elements. By retrieving content from these two perspectives, we enable the language model to effectively mimic the process of capturing both neighboring and broader contexts when processing aggregated information. This dual approach ensures a richer understanding of the data, enhancing the model's ability to generate more accurate and contextually relevant outcomes.

---

**Algorithm 1** Hash-based Jaccard Similarity Ranking

---

**Input:** $K$-hop neighboring ID sequences, each containing $V$ nodes, processed by $R$ independent MinHash functions $\{h_1, h_2, \ldots, h_R\}$, each with a range of $B$.
**Output:** ranked $score$ of node IDs.
**for** node $v \in V$ **do**
  **for** $hop = 1 \to k$ **do**
    $\mathcal{N}^k(v) = \text{GetNeighbors}(v,k)$
  **end for**
**end for**
**Initialize:** $H_v \leftarrow \mathbb{M}^{V \times (R \cdot K^2)}$
**Initialize:** $C_v \leftarrow \mathbb{M}^{V \times (R \cdot K^2)}$
**for** $s \in \mathcal{N}^k(v)$ **do**

**for** $hop = 1 \to k$, $c_v \leftarrow []$ **do**
  **for** $r = 1 \to R$ **do**
    Append $h_r(s)$ $K$ times to $H_v$
    Append $h_r(s)$ to $c_v$
  **end for**
  Append $c_v$ $K$ times to $C_v$
**end for**
**end for**
**for** $s \in \mathcal{N}^k(v)$ **do**
  $Score_v = Sum_v[h_v == C_v]$
  Rank $Score_v$ in descending order
**end for**
**return** $Score$

---

## 3 EXPERIMENTS

We conduct extensive experiments on three real-life datasets. Our study aims to address the following research questions: **Q1:** Can SKETCH achieve superior prediction performance than current state-of-the-art frameworks without utilizing graph neural networks? **Q2:** How effective are semantic retrieval and structure retrieval modules in selecting augmented textual information from other nodes, and how do they perform under different scenarios? **Q3:** What is SKETCH's sensitivity to its hyperparameters, and how is the efficiency of hash simulation compared to the standard method?

**Implementation Details.** We use a server with six 24 GB NVidia RTX 3090 GPUs. Our method utilizes the Adam optimizer with a learning rate of 0.001 and incorporates early stopping based on validation set accuracy. Main experiments are evaluated by the prediction accuracy of the testing set, with performance results on the validation dataset also included. Hyperparameters for length $l$ and $k$ hops are fine-tuned using a grid search to select the optimal values for each dataset. All baseline experiments follow the design outlined in their respective articles to ensure fairness. For detailed descriptions of the datasets and further explanations of the baselines, please refer to the appendix A.

### 3.1 MAIN RESULTS

The comparison of prediction performance across three datasets between SKETCH and other baseline methods is presented in Table 1. The best result for each baseline group has been highlighted by underlying. We have categorized all benchmarks into four groups: (1) traditional fine-tuned BERT-based models with GNN, (2) recent efficient parameter fine-tuning methods for LLMs, (3) leveraging powerful chat models like GPT-4 through in-context learning, and (4) existing tailored approaches using various techniques. Specifically, the traditional BERT-based method yields reasonable results thanks to the flexibility of fine-tuned embeddings. In today's landscape of large language models, larger sizes indeed bring about improved quality. An interesting finding is that using prompts to guide LLMs in classification is unsatisfactory. Specifically, the Llama2 models struggle to follow instructions and often generate irrelevant content. It's quite sensitive to the phrasing of prompts, and minor changes in words can lead to significantly different outputs. GPT-4 models perform better but remain inferior to specialized trained models. SKETCH surpasses all baselines in overall accuracy,

reaching an average improvement of 1.2%. This achievement is attributed to the use of informative retrieved text and a long-context model, with the combined corpus enhancing performance from both semantic and structural perspectives. We have chosen two distinct language models as the backbone of our framework: Nomic, which has 137 million parameters, and Llama-3, with 8 billion parameters. The larger Llama-3 model demonstrates higher performance but demands significantly more time and memory. Training Nomic takes less than one hour per epoch, while Llama-3 requires over 9 hours on the ACM dataset. This trade-off highlights the need to consider both aspects, and researchers can select the appropriate option according to their real-life situations.

Table 1: Performance comparison among state-of-the-art baselines on three benchmark datasets.

| NLP Models | GNNs | ACM | | Wikipedia | | Amazon | |
|---|---|---|---|---|---|---|---|
| | | Val-Acc. | Test-Acc. | Val-Acc. | Test-Acc. | Val-Acc. | Test-Acc. |
| **Fine-tuned LMs +/- GNNs** | | | | | | | |
| Bert | - | 74.4 | 73.2 | 69.5 | 68.8 | 86.2 | 87.0 |
| | GCN | 77.6 | 77.1 | 69.4 | 68.4 | 92.3 | 92.8 |
| | GAT | 77.9 | 78.0 | 70.5 | 69.8 | 92.5 | 92.4 |
| | GraphSAGE | 77.3 | 76.8 | 73.1 | 72.7 | 92.0 | 92.3 |
| Roberta | - | 78.1 | 76.6 | 67.8 | 68.1 | 84.9 | 85.9 |
| | GCN | 80.1 | 79.4 | 68.5 | 68.0 | 92.3 | 92.5 |
| | GAT | 79.7 | 78.9 | 70.1 | 71.0 | 92.5 | 92.4 |
| | GraphSAGE | 78.5 | 78.3 | 72.7 | 72.1 | 92.2 | 92.1 |
| **Fine-tuned Large Language Models +/- GNNs** | | | | | | | |
| Llama3-8b | - | 80.7 | 80.6 | 71.9 | 71.2 | 92.0 | 91.6 |
| Llama3-8b | GraphSAGE | 82.0 | 81.3 | 72.8 | 73.0 | 93.1 | 92.8 |
| **Large Language Models** | | | | | | | |
| Llama2-7b | | - | 20.8 | - | 41.3 | - | 53.4 |
| Llama2-13b | | - | 58.9 | - | 48.9 | - | 57.6 |
| GPT-3.5 | | - | 54.3 | - | 61.8 | - | 49.1 |
| GPT-4 | | - | 67.5 | - | 60.9 | - | 40.3 |
| **Tailored Frameworks For TAG** | | | | | | | |
| MPAD | | 74.9 | 74.6 | 70.3 | 70.4 | 88.2 | 88.0 |
| GLEM | | 76.1 | 76.2 | 69.8 | 70.2 | 88.9 | 88.7 |
| LLAGA | | 77.2 | 77.5 | 71.7 | 72.0 | 90.1 | 90.8 |
| GraphFormers | | 75.3 | 75.1 | 66.8 | 67.5 | 85.6 | 86.4 |
| InstructGLM | | 76.7 | 75.6 | 72.2 | 71.2 | 94.2 | 94.0 |
| Ours (Nomic) | | 81.4 | 81.1 | **74.1** | **73.6** | 93.3 | 93.5 |
| Ours (Llama3-8b) | | **82.7** | **82.3** | 73.3 | 73.4 | **94.1** | **94.7** |

## 3.2 Effectiveness of Retrieval Modules

To evaluate the effectiveness of the retrieved content in improving performance, we analyze accuracy under various conditions. The baseline benchmark uses only the text from the anchor node. We then conduct the following experiments: (1) randomly incorporating text from other nodes in the graph, (2) retrieving only semantically similar content, (3) retrieving structurally similar content with three variants, and (4) testing our proposed SKETCH model. The analysis results are presented in the table 2. Our findings highlight several important insights regarding the impact of content addition on performance. Firstly, our research shows that randomly incorporating unrelated material into the original text does not enhance performance; in fact, it negatively affects the overall results. This underlines the critical importance of retrieving information that is relevant to the context. While we observed that texts with global similarities can provide some level of positive influence, they are still less effective compared to one-hop neighboring texts. This suggests that having connected edges serves as a valuable reference for anchor nodes, strengthening their relevance. Furthermore, our experiments demonstrate that arbitrarily extending the range of neighbors does not provide additional benefits, which further confirms the effectiveness of our hash-based similarity ranking scheme.

Table 2: Comparisons across various settings. *k-hop* indicates the range of hops used to retrieve neighbors, while *Selection* refers to either random sampling or similarity ranking as the criterion.

| Variant | Retrieval Strategy | | | | | | |
|---|---|---|---|---|---|---|---|
| | Original | Shuffled | Semantic | One-hop | Multi-hops | Combined | Ours |
| Semantic | × | × | ✓ | × | × | ✓ | ✓ |
| Structural | × | × | × | ✓ | ✓ | ✓ | ✓ |
| K-hop | - | - | - | 1 | 3 | 3 | 3 |
| Selection | - | Random | Rank | Random | Random | Random | Rank |
| Accuracy | 78.0 | 75.6 | 78.4 | 80.6 | 79.7 | 80.3 | 81.4 |

## 3.3 HYPERPARAMETER STUDY AND EFFICIENCY COMPARISON

Our framework's complexity mainly depends on three factors: the length of the tokenized sequence $L$ for each concatenated paragraph and the weights of the semantic and structural retrieval modules. The ablation study evaluates classification accuracy across various configurations of these factors. Figure 4 indicates that while longer contexts can be beneficial, excessively lengthy sequences may diminish overall understanding and introduce noise that confuses the model. Additionally, for single-machine users, longer texts lead to smaller batch sizes, which can further decrease performance. Another finding is that increasing the weights of structure ranking generally enhances performance, emphasizing the importance and effectiveness of our intersection-based retrieval strategy. Our model shows no significant drop in performance, indicating it is not excessively sensitive to hyperparameters. We also compare the time taken by our hash-based simulation with the standard computing method. Our experiments indicate that adding extra hops does not lead to a linear increase in time, as illustrated in figure 5. In contrast, the standard method requires exponentially more time due to the complex cross-combination of multiple hops. This distinction highlights the efficiency of our approach, which maintains time consumption even while considering extra ranges.

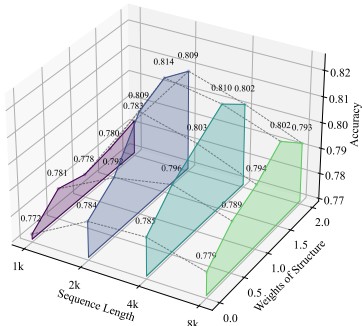

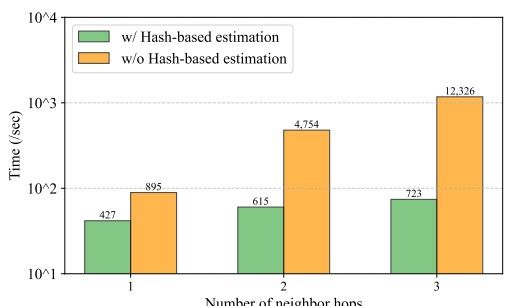

Figure 4: Effects of hyperparameters on the performance, showing the impact of sequence length and structural weights.

Figure 5: Comparison of computational efficiency between hash-based and standard one. The time metrics have been logarithmically processed.

## 4 RELATED WORK

### 4.1 LLMS FOR GRAPHS

Text-attributed graphs (TAGs) possess long texts as node attributes, allowing language models (i.e.,Bert (Devlin et al., 2019), GPT2 (Radford et al., 2019)) to significantly enhance text learning. Large language models(LLMs) have shown increasingly powerful performance in text understanding, especially large-scale texts. Therefore many recent researches apply LLMs to downstream tasks of TAGs (Li et al., 2024b), such as classification and link prediction (Tan et al.), reasoning (Luo et al., 2024), graph generation (Yao et al., 2024) etc. To further enhance the performance, there are many techniques, such as fine-tuning (Dernbach et al., 2024), instruction-tuning (Chung et al., 2024) and prompt design (Guo et al., 2023; 2024). In graph-related tasks, prompts combin-

ing graph descriptions are common. For example, TAPE (He et al., 2024) and SimCSE (Li et al., 2024a) concatenates generated relevant information or similar neighbors to provide additional information for representation learning. However, these methods are limited by computation sources or the limited sequence length of LLMs (Lee et al., 2024). Additionally, some research proposes complex pipelines to fine-tune LLMs for graph learning. Pan et al. (2024) aligns the student model and interpreter model from semantics, structures, and prediction probabilities. Moreover, Guo et al. (2024) and Tang et al. (2024) apply instruction-tuning, one by refining the graph structure, and the other with text-structure alignment. Besides, many deep learning methods are enforced, like contrastive leaning (Zhang et al., 2024a) and ensembling (Zhang et al., 2024b). Although these methods attempt to leverage graph structures in LLMs, they focus on neighborhood nodes but are weak in utilizing topological structures. Many other models utilize the generation ability of LLMs under particular settings, including label-free tasks Chen et al. (2024b), few-shot and zero-shot learning situations Liu et al. (2023). However, these works fall short of exploiting the advantages of LLMs, limiting capability in contextual understanding.

## 4.2 LLMs with GNNs

To bridge semantics understanding and graph structures, cascaded LLMs with graph neural networks (GNNs) have emerged in classification tasks. Currently, related reviews categorize models into three types: LLM-as-enhancer, LLM-as-predictor, and LLM-GNN alignment (Li et al., 2024b). TAPE (He et al., 2024), a typical example of the first type, generates supplementary contexts. Similarly, Fang et al. (2024) not only applies multiple language models to augment features with the mixture of prompt experts but also employs the edge modifier to adjust neighbor weights during graph learning. While RoSE (Seo et al., 2024) decomposes relations by generator and discriminator by LLMs to provide structure information for multi-relational GNNs. In contrast, the second model, i.e., Dr.E (Liu et al., 2024), transfers output from GNNs to the language decoder to decompose into features, edges, and labels. These two types have challenges of losing information or misunderstanding during transformation. Except for these two cascaded designs, there are nested integrations (Yang et al., 2024). ENGINE (Zhu et al., 2024) embeds a side structure of LLMs by simple neural network layers as ladders, and LinguGKD (Hu et al., 2024) aligns both in each layer by leveraging contrastive distillation loss. This synergy combines neighborhood aggregation and semantic learning, but they find it difficult to address co-training problems or alignment errors.

## 5 Conclusion

Inspired by the potential of language models to manage text-attributed graphs, we introduce SKETCH, a new approach that simulates graph propagation through weighted token learning from selected node subsets. Our framework enhances the selection of informative data by retrieving nodes that are both semantically and structurally similar, thus enriching the textual information. To reduce the computational complexity of node intersection calculations, we implement a novel hash-based estimation technique. Extensive experiments demonstrate that our model outperforms all baseline methods while requiring less time and memory, eliminating the need for GNNs. Additionally, we conduct a thorough analysis of various settings to identify key components that positively impact text-attributed graph learning. Our proposed framework demonstrates significant potential and offers valuable insights into the integration of large language models within graph-related applications.

## 6 Reproducibility

To ensure the reproducibility and verifiability of our results and models, we provide a complete codebase and step-by-step usage instructions. Specifically, (1) the results and related analysis reported in the paper are only a summary of those available in the code; (2) the implementation of SKETCH and all baseline models are accessible; (3) detailed information and settings about main parameters are publicly available; (4) information regarding hyperparameter tuning and training times are included; (5) the hardware used is also disclosed; (6) in a fixed environment (i.e., with the same hardware and software versions), most results are bitwise reproducible.

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

## A APPENDIX

### A.1 DATASET

We assess the performance of SKTECH using the following three datasets: ACM, Wikipedia, and Amazon. All of these three datasets are manually constructed using the raw corpus and corresponding descriptions. For each dataset, we divide the labels into training, validation, and testing sets. Statistics of these three datasets are shown in Table 3.

**Wikipedia.** The raw data consists of UTF-8 encoded text from Wikipedia articles[1]. We extract the main content of each article as document $d_v$, which includes hyperlinked words. A directed graph is constructed using the hyperlink relationships between articles. The categories mentioned in the *list of reference tables* are assigned as labels to the nodes.

**ACM.** This dataset uses 48,579 papers from the Association for Computing Machinery (ACM) as instances Tang et al. (2008). The paper abstracts serve as the document $d_v$ for the nodes, and a directed graph is constructed using the citation links. The instances are collected from nine distinct domains, such as Artificial Intelligence, Data Mining, and Machine Learning, which are employed as labels.

**Amazon.**The dataset comprises product reviews and metadata from Amazon He & McAuley (2016). We construct the graph based on the browsing history, with each node $v$ representing the textual description of the products denoted as $s_v$.

Table 3: Statistics of datasets in our experiment.

| Datasets | #nodes | #edges | #classes |
|----------|--------|--------|----------|
| ACM | 48,579 | 193,034 | 9 |
| Wiki | 36,501 | 1,190,369 | 10 |
| Amazon | 50,000 | 632,802 | 7 |

### A.2 BASELINES

As our study focuses on integrating the corpus proceeding with the graph network, we adopt a variety of popular approaches in these two domains, i.e. text-embedding modules and GNN encoders. We made a cross combination of frontier methods in each field. Here are the introductions of each method:

- GCN Kipf & Welling (2017) aggregates information from neighboring nodes by summing over neighbors' representations.
- GraphSAGE Hamilton et al. (2017) samples and aggregates features from the neighborhood for inductive graph learning.
- GAT Brody et al. (2022) introduces a dynamic graph attention mechanism, leveraging attention layers to learn the weights of neighboring features.
- Bag of Words (BoW) Zhang et al. (2010) describes the occurrence of words within a document and its size can be flexibly decided by the frequencies of different words.
- MPAD Nikolentzos et al. (2020) represents corpus as networks based on word co-occurrence and applies a message-passing framework to draw the information from the graph.
- Fine-tuning a language model (LM-tune) allows for training on target texts to make the model more adept at performing the specific task.
- GLEM framework Zhao et al. (2023) iteratively updates the language model and graph neural network (GNN).
- GraphFormers Yang et al. (2021) integrate GNN components with transformer modules for joint training rather than a cascaded approach.

---

[1]http://www.mattmahoney.net/dc/textdata

- LLAGA Chen et al. (2024a) effectively integrates LLM capabilities to manage the complexities of graph-structured data.

- Llama Touvron et al. (2023) is a family of large-scale language models that are designed to understand and generate human-like text across various tasks.

- GPT Floridi & Chiriatti (2020) is a set of state-of-the-art language processing AI models.

- InstructGLM Ye et al. (2023) employs natural language to characterize the multi-scale geometric structure of graphs and fine-tunes a large language model (LLM) for graph tasks.

