# OpenReview forum: "Large Language Models based Graph Convolution for Text-Attributed Networks"
_ICLR.cc/2025/Conference — Submitted to ICLR 2025_

### Official Review · Reviewer_sKWC · 2024-10-21

**Soundness:** 2
**Presentation:** 3
**Contribution:** 2
**Rating:** 5
**Confidence:** 4

**Summary:**

This paper proposed Semantic Knowledge and Structural Enrichment framework (SKETCH) to extract the semantic and structural related information from the graph to help graph understanding and reasoning. The conducted experiments show that SKETCH could enhance the model's performance on three graph datasets.

**Strengths:**

- The proposed method is well motivated and easy to follow
- SKETCH givens a new perspective to integrate LLMs and the graph task
- The writing and presentation of this paper is clear

**Weaknesses:**

- The evaluation is only limited to 3 datasets with less than 10 classes. InstcurtGLM [1] was evaluated on Ogn-arxiv, while GraphFormers [2] was evaluated on Product, DBLP and Wiki.
- The improvement from Llama3-8b+GraphSAGE scenario is marginal.
- SKETCH requires extensive hyper-parameters tuning compared to existing graph based methods (such as Llama3-8b+GraphSAGE).

[1] Ye, Ruosong, et al. "Natural language is all a graph needs." arXiv preprint arXiv:2308.07134 4.5 (2023): 7.

[2] Yang, Junhan, et al. "Graphformers: Gnn-nested transformers for representation learning on textual graph." Advances in Neural Information Processing Systems 34 (2021): 28798-28810.

**Questions:**

- Could you release the train/test/val splits of the three datasets? I haven't found it in Appendix A.1 and main text.
- Could you provide more explanation on the claim that SKETCH requires fewer computational resources than other baselines?

---

> ### Author Response · Authors · 2024-11-16
> **Response to reviewer sKWC**
>
> Dear Area Chair and Reviewers:
>
> Thank you for your commitment to reviewing our paper. Below are our detailed responses to your inquiries:
>
> **[W1] Datasets**
>
> Our three datasets contain commercial, citation, and scientific terms, all constructed from real-life data. We have noticed that some high-performing frameworks on the leaderboard. But they do not train from scratch. Instead, they integrate embeddings from previously established frameworks. In fact, some of the top models even jointly combine features from four different methods, which could make the evaluation unfair. To address this, we replicated these methods on our own datasets and provided a reasonable comparison.
>
>
> **[W2] Marginal Improvement over LLM + GNN -- One key advantage of our study is to get rid of GNN and it's purely trained by LLMs.**
>
> We appreciate the feedback regarding the significance of our improvement. We would like to emphasize that a 1% performance increase can lead to substantial real-world impacts. Additionally, our method shows even greater improvements on the Amazon and Wiki datasets, highlighting its superior generalization capability and adaptability in real-life applications. Furthermore, our approach is simpler than those that rely on Graph Neural Networks (GNNs). Our model is more lightweight and time-efficient compared to methods that rely on Graph Neural Networks (GNNs), which makes it easier to implement and deploy. The training time for Llama-3 takes over a day on a single GPU, while Nomic can complete it in just one hour. Additionally, GNNs require batch training when dealing with particularly large graphs, which significantly increases computation time.
>
> **[W3] Hyperparameters tuning -- Our framework is not hyperparameter sensitive and we require fewer computational resources than LLM + GNN.**
>
> Our model is not sensitive to hyperparameters, as we have demonstrated in the paper. We only require a single weight to control ranking, and in most cases, the default settings of a sequence length of 2000 and a weight of 1.5 are sufficient.
>
> **[Q1] Split ratio of datasets -- We adopted a 6:1:1 ratio.**
>
> Thank you for your kind reminder regarding the train/test/validation split. We have used a 6:1:1 ratio for the splits, and we will update this information in the appendix of our paper. We also experimented with an 8:1:1 ratio, and found that the comparisons between different methods remained consistent across both settings.
>
> **[Q2] Efficiency of SKETCH**
>
> Thank you for raising this valuable question regarding how our framework can use fewer computational resources. In fact, this is one of the main motivations behind our work. Traditional TAG methods typically involve first using NLP embeddings to obtain text representations and then employing Graph Neural Networks (GNNs) to learn the graph structure. However, GNNs can be very resource-intensive, especially when dealing with large graphs, leading to batch training times that can extend to several hours. Some recent methods, like GLEM and graphformers, attempt to combine these two stages through various algorithms. In contrast, our approach focuses on using only LLMs for TAG, as LLMs facilitate weighted learning for each token. This necessitates finding an appropriate corpus for the LLM, enabling it to perform weighted learning on the text associated with each node. To address this, we introduced a module for the retrieval of semantic and structural content. While some methods leverage in-context learning approaches, such as using GPT-4 for prediction and content generation, these methods often require significant budget and time. In contrast, our SKETCH model, when using Nomic, only requires a little over an hour of training, making it far more efficient than other approaches.

---

> > ### Comment · Reviewer_sKWC · 2024-11-25
> >
> > I appreciate the author's detailed rebuttal. With the clarification on efficiency and experiments, I have raised my score accordingly.

---

### Official Review · Reviewer_vLaZ · 2024-11-02

**Soundness:** 2
**Presentation:** 2
**Contribution:** 2
**Rating:** 3
**Confidence:** 5

**Summary:**

The paper introduces SKETCH, a novel framework for handling text-attributed graphs (TAGs) based on retrieval-augmented generation, enhancing large language models (LLMs) for TAG-related tasks.

**Strengths:**

1. The retrieval-based approach offers a fresh perspective on handling TAGs.

2. The model leverages hash-based similarity estimation to reduce computational costs in multi-hop similarity estimation.

**Weaknesses:**

1. Work on graph retrieval-augmented generation, which is closely related to the topic of this paper, is not discussed.

2. Lack of implementation details such as hyparameter searching space, prompts used, etc.

**Questions:**

1. The foundational work of Retrieval-Augmented Generation (RAG) [1] is not cited. Given that the primary contribution of this paper lies in graph retrieval-augmented generation, it is crucial for the authors to provide a comprehensive discussion of significant prior works [2-4] in related fields.

2. Only texts in documents (nodes) are used, and connections (graph structure) between texts are not considered in the generation phase. The authors presents several drawbacks in TAG modeling, such as "the text representations and graph structure are trained independently from their respective aspects, potentially resulting in sub-optimal integration between the two modalities" and "the separate processing stages do not take into account the simultaneous optimization of the two data types, resulting in information loss and reduced robustness". Could the authors clarify how SKETCH addresses these challenges?

3. The paper lacks implementation details and accessible code. How do authors fine-tune LLMs? What is train / val / test split? What is the searching space for each hyperparameter, e.g., $k$-hop? The reproducibility claims in the article are not convincing. The claim that "the results and related analysis reported in the paper are only a summary of those available in the code" is ambiguous.

4. What is $G$ in $G = R_{sum} + R_{struct}$?

5. It is fair to use frozen / fine-tuned LLMs as baseline. However, comparing the proposed model with tailored TAG models that do not utilize external knowledge bases may be unfair. Why not include RAG-based approaches for TAGs?

6. What is external knowledge data used for each dataset?

7. What are promps used for proposed model (SKTECH)? Are the prompts employed for the LLM baseline the same as those used for SKETCH?

8. Why SKTECH with Nomic (127M parameters) perform better than SKTECH with Llama3 (8B parameters) on Wikipedia when Nomic has much fewer parameters?

---
[1] Lewis, Patrick, et al. "Retrieval-augmented generation for knowledge-intensive nlp tasks." Advances in Neural Information Processing Systems 33 (2020): 9459-9474.

[2] He, Xiaoxin, et al. "G-retriever: Retrieval-augmented generation for textual graph understanding and question answering." arXiv preprint arXiv:2402.07630 (2024).

[3] Hu, Yuntong, et al. "GRAG: Graph Retrieval-Augmented Generation." arXiv preprint arXiv:2405.16506 (2024).

[4] Edge, Darren, et al. "From local to global: A graph rag approach to query-focused summarization." arXiv preprint arXiv:2404.16130 (2024).

---

> ### Author Response · Authors · 2024-11-20
> **Response to reviewer vLaZ**
>
> Dear Area Chair and Reviewers:
>
> Thanks for your efforts and we would like to answer the questions you raised.
>
> **[W1] Not include RAG papers -- We are not doing GraphRAG but inspired by the idea ofretrieval of useful content.**
>
> We would like to clarify that our article is not based on GraphRAG. While Retrieval-Augmented Generation (RAG) leverages in-context learning and the powerful external knowledge of large language models (LLMs), we found that its classification performance on certain datasets is not satisfactory. Our approach focuses on training a customized long-context model tailored to specific downstream tasks.
>
> The main advantage of our method over traditional approaches is that we do not rely on Graph Neural Networks (GNNs) to aggregate information from the graph structure. Instead, we are motivated by retrieval-based principles, which allow us to select useful content to enhance the understanding of the trained model.
>
> **[W2] Details of experiment settings.**
>
> Thank you for raising the question regarding the setup. We would like to clarify the framework's process. For each anchor node, we retrieve relevant items based on structural and semantic similarity, and we rank their relatedness using a weighted ranking approach. The parameter K controls the trade-off between these two aspects, while the sequence length determines the final input content length.
>
> In our hyperparameter study, we demonstrated the impact of these factors on the results and found that our model is not sensitive to hyperparameters, making it robust. Regarding prompts, since we are not utilizing in-context learning, we simply concatenate useful content rather than designing specific prompts.
>
> **[Q1] Discussions about RAG. -- We will update it in the related work session.**
>
> We have indeed drawn inspiration from the concept of retrieving useful content in RAG; however, we want to emphasize that our framework is primarily based on a training approach. Thank you for your suggestions and we will include all the papers you mentioned in the related work section.
>
> **[Q2] Clarification of how SKETCH addresses certain challenges.**
>
> Thank you for your thoughtful question regarding the challenges in traditional TAG modeling. In conventional approaches, TAG models first train language models to embed the text of each node and then feed these embeddings into a Graph Neural Network (GNN). This two-stage method often leads to sub-optimal results due to the lack of an end-to-end optimization process, which can introduce biases towards each aspect independently.
>
> In contrast, our method, SKETCH, utilizes a Large Language Model (LLM) to simultaneously learn both text and graph representations. By retrieving related nodes, the LLM can provide weighted learning for each token, enabling a more unified approach that integrates text and structural information effectively. This allows SKETCH to better capture the relationships between nodes and optimize performance for downstream tasks more cohesively.
>
> **[Q3] Experimental Settings.**
>
> Thank you for your inquiry regarding our experimental setup. To clarify, our dataset is split into training, validation, and test sets following a 6:1:1 ratio. We have also experimented with alternative splits, such as 8:1:1, and found that the results remained consistent across these configurations.
>
> Regarding the K-hop results, we provide a thorough analysis in Section 3.2, where we explore various configurations, including 1-hop and 3-hop scenarios, as well as different retrieval strategies. This detailed evaluation demonstrates the effectiveness of each module in our framework.
>
> For training our LLM, we utilized the classic Low-Rank Adaptation (LoRA) tuning method, which allows us to efficiently fine-tune the model parameters while maintaining computational efficiency.
>
> **[Q4] What is G = R_sem + w * R_struct? -- It represents the overall score of both semantic and structural relatedness.**
>
> In this context, G represents the overall score for the relatedness of each node. During the retrieval process, we focus on both semantic and structural retrieval aspects, and this equation serves to combine their rankings into a single cohesive score.
>
> Here, R_sem denotes the semantic relatedness score, while R_struct represents the structural similarity score. The parameter w controls the weight given to the structural similarity component. We typically set w to values around 1.5, and our ablation studies have shown that this hyperparameter is relatively robust, resulting in minimal fluctuations in performance.

---

> > ### Author Response · Authors · 2024-11-20
> > **Additional Reply**
> >
> > **[Q5] RAG-based baselins.**
> >
> > Thank you for your question regarding the comparison with other RAG baselines. In fact, our baseline model, InstructGLM, is built upon a RAG framework. It leverages prompt design to utilize GPT-4 for providing explanations and predictions related to anchor nodes, which enhances the training process and ultimately contributes to the final predictions. This approach benefits from external knowledge and the understanding capabilities of the LLM.
> >
> > However, our method, SKETCH, demonstrates superior performance compared to this baseline. We have a concern that while RAG shows promising results on citation graphs and similar TAGs due to the presence of academic papers in the LLM's training data, it may significantly underperform in more specific domains where such domain-specific knowledge is less prevalent. In this case, training is still a primary choice.
> >
> > **[Q6] External knowledge.**
> >
> > Large Language Models (LLMs) like GPT-4 benefit significantly from having been trained on extensive datasets, which include substantial external knowledge from sources like citation graphs and Wikipedia. These sources provide a wealth of information that enhances the model's ability to understand and generate relevant content, as it is likely that the model has encountered similar information during training.
> >
> > **[Q7] What are prompts we used? -- We concatenate retrieved content.**
> >
> > It's important to clarify that we are not operating as a traditional RAG model. Instead, our method focuses on retrieving relevant content and contatenating it together to provide richer contextual information for training.
> >
> > This approach allows us to enhance the input data without needing particularly complex or "tricky" prompts. Our emphasis is on utilizing the retrieved content effectively to improve the overall performance of our model. By simply concatenating the retrieved information, we can ensure that the model has access to a broader range of relevant context during training.
> >
> > **[Q8] Why nomic performs better. -- Larger models do not always mean better performance.**
> >
> > Thank you for your question about why Nomic sometimes yields better results. One key factor is that Nomic employs a customized long-context model, which is specifically designed to handle larger input sequences effectively. However, it's important to note that a larger model does not inherently guarantee better performance.
> >
> > In our experiments, we trained all models under the same GPU resources, which means that Nomic may effectively utilize a larger batch size during training. This capability can lead to improved learning dynamics and better generalization.
> >
> > Additionally, it’s crucial to recognize that, unlike typical text generation tasks where model size is often emphasized, selecting the appropriate model for the specific task at hand can be more critical. The architecture and training strategy tailored to the dataset and objectives can significantly influence performance.

---

> ### Author Response · Authors · 2024-11-29
> **Could you please share your comments?**
>
> Dear reviewers,
>
> Thanks for your reviews. I understand you have a busy schedule, but I would appreciate any additional comments on my replies. If you need further clarification or have questions, please let me know. Your feedback is valuable, and we are looking forward to your response. Thank you again!
>
> Yours,
>
> Authors

---

### Official Review · Reviewer_jsyY · 2024-11-03

**Soundness:** 2
**Presentation:** 2
**Contribution:** 2
**Rating:** 5
**Confidence:** 5

**Summary:**

The author has introduced a new method that combines a pre-trained language model (PLM) with a graph heuristic (Common Neighbor) for semi-supervised node classification. In this approach, the PLM generates semantic proximity by incorporating a weighted sum of token-level embeddings. This semantic information is then fused with local graph structure, such as the common neighbor heuristic, by weighting the local connections according to their semantic proximity.

**Strengths:**

1. Due to the missing semantic information during the message-passing process, the author has proposed a fused framework based on LM and graph heuristics, which is easily scalable.
2. The author has conducted extensive experiments to demonstrate the performance improvement and computational efficiency.

**Weaknesses:**

1. When introducing the background of GCN and RAG, some crucial papers are not cited. For instance, the structure of text-attributed graphs encompasses textual information from various nodes. Inspired by concepts from Retrieval-Augmented Generation (RAG) \cite{one in NLP}{one in graph}{one in tag}, we propose integrating an additional corpus during the training process.

2. The paper requires additional revision for better and more fluent logical flow. For example, GNNs primarily depend on node-level aggregation via graph convolutions to compute weighted sums of neighboring features. While effective, this method may overlook the rich semantic nuances in textual data. Incorporating such nuances enables a more granular understanding of the relationships between tokens, leading to improved flexibility and adaptability.

3. The method is not well presented. In the section on aggregated learning of retrieved content, a weighted sum of semantic and structural proximity is introduced, but the difference from graph convolution is not carefully studied or justified.

4. Empirical Demonstration: The results are reported without running on 5-10 random seeds or multiple data splits.

5. It is not clear how to calculate semantic proximity and structural proximity for a node classification task.

**Questions:**

1. Is the method reproducible? Please provide your repository if possible.
2. Is the GCN-generated embedding also leveraged in the method? Has the author carefully considered the differences between the proposed method and GCN, or should these differences be discussed further?
3. The title is **LARGE LANGUAGE MODELS BASED GRAPH CONVOLUTION FOR TEXT-ATTRIBUTED NETWORKS**. Does this imply that feature aggregation based on a weighting mechanism quantified by semantic and structural proximity is a better way to present it?
4. "While effective, this method may miss the rich semantic nuances in textual data." This is probably a crucial starting point. Are there any references or experiments to support this claim?

---

> ### Author Response · Authors · 2024-11-23
> **Response to reviewer jsyY**
>
> Dear Area Chair and Reviewers:
>      Thank you for your commments and suggestions about our paper. Next are our detailed responses and explanations about each comment:
>
> **[W1]Citation of Crucial Papers**
>
> Thank you for your observation regarding the citations of significant papers related to GCN and RAG. The absence of citations was considering, as our semantic similarity approach is inspired by the RAG concept. However, we appreciate your valuable suggestion to include relevant literature on NLP and RAG.  Additionally, we recognize the need to add necessary citations in the methodology section to help readers better understand related research.
>
> **[W2]Logical Flow and Clarity**
>
> We appreciate your feedback on the logical flow and clarity of our paper. We recognize the need for a more straightforward and coherent methodology to enhance understanding. Thank you for this valuable suggestion; we will prioritize these writing improvements in our upcoming revision.
>
> **[W3]Presentation of the Method**
>
> Thank you for highlighting the need to improve the presentation of our method. We will revise the aggregated learning section to better clarify the differences between the weighted sum of semantic and structural proximity and traditional graph convolutions. First, I’d like to outline the differences here. Although our approach mimics GCN, it differs in that we aggregate texts instead of node features or embeddings. This allows us to fine-tune language models or large language models without being constrained by graph sizes and scales, whereas using embeddings would require more complex handling. Additionally, our model simulates weights through scores to determine the rank in which texts are concatenated. However, our approach has limitations, such as the lack of dynamically changing weights and the capacity for multiple aggregation processes.
>
> **[W4]Empirical Demonstration**
>
> We appreciate your suggestion regarding the empirical demonstration of our results. We have collected average results on several random seeds and our performance is very stable under different seeds. About the dataset train/test/validation split, we have used a 6:1:1 ratio for the splits, and we will update this information in the appendix of our paper. We also experimented with an 8:1:1 ratio, and found that the comparisons between different methods remained consistent across both settings.
>
> **[W5]Calculation of Proximity Measures**
>
> Thank you for your feedback regarding the calculation of semantic and structural proximity for the node classification task. We realize that our explanation of these concepts was not sufficiently clear. For node semantic similarity, we use the FAISS tool to search for similar nodes across the entire corpus based on cosine similarity. For structural similarity, we apply a Minhash-based method to compute Jaccard similarity. Additionally, the figures provide a general overview of these two measures. We appreciate any further suggestions or questions you may have.

---

> > ### Author Response · Authors · 2024-11-23
> > **Additional reply**
> >
> > **[Q1]Reproducibility**
> >
> > Thank you to the reviewer for raising the important issue of reproducibility. This is a key aspect of our research. Our current plan is to share our methods and code after the paper is published.
> >
> > **[Q2]Comparison with GCN**
> >
> > Thank you for your valuable feedback. Regarding the GCN (Graph Convolutional Network) based embedding, we did not directly utilize this method in our approach. Instead, our proposed method serves as a mimic of GCN, where we do not use GCN embeddings; rather, we process aggregated texts based on semantic and structural nodes to simulate how GCN aggregates neighbors' embeddings. In our approach, the selection of texts is influenced by scoring similar to weights, which determines the importance of the nodes being aggregated. Your question is indeed thought-provoking, and it highlights the need for us to further analyze the distinctions between our method and GCN. We will seriously consider this suggestion in the revision.
> >
> > **[Q3]Feature aggregation**
> >
> > Your suggestion makes a lot of sense. This approach could indeed utilize weights based on feature representations, which aligns more closely with GCN. However, during implementation, especially with larger graphs, there are limitations to your idea. Processing the nodes in each batch requires providing corresponding semantic structural node features, which is quite challenging for language models (LMs). We simulate weights through scores and concatenate texts, which is a simpler operation that can be handled by LMs with longer context capabilities. Additionally, the node search process is independent, which helps save fine-tuning time for the LMs.
> >
> > **[Q4]Rich semantic nuances**
> >
> > Thank you for your valuable questions. This sentence aims to claim that language models have a better understanding of texts on text-attributed graphs and explore deeper semantics of texts. Before we use GNN to learn node embeddings, we need to process texts as node features by methods such as word2vec or IF-IDF. The results in reference[1] Table 2 have shown that in most circumstances, GNN-as-predictor is not better than finetuning language models.
> >
> > [1] J. Yu, Y. Ren, C. Gong, J. Tan, X. Li, and X. Zhang, “Empower Text-Attributed Graphs Learning with Large Language Models (LLMs),” Oct. 15, 2023, *arXiv*: arXiv:2310.09872. Accessed: Sep. 29, 2024. [Online]. Available: http://arxiv.org/abs/2310.09872

---

> ### Author Response · Authors · 2024-11-29
> **Could you please share your comments?**
>
> Dear reviewers,
>
> Thanks for your reviews. I understand you have a busy schedule, but I would appreciate any additional comments on my replies. If you need further clarification or have questions, please let me know. Your feedback is valuable, and we are looking forward to your response. Thank you again!
>
> Yours,
>
> Authors

---

### Official Review · Reviewer_L5Yj · 2024-11-03

**Soundness:** 3
**Presentation:** 3
**Contribution:** 3
**Rating:** 6
**Confidence:** 4

**Summary:**

This paper investigates an interesting problem of leveraging LLM for learning on text-attributed graphs. The authors propose a method called SKETCH which adapts LLM for graphs by retrieving both structural and semantic information. To be specific, the semantic-based retrieval is built upon some off-the-shelf pretrained retrievers, and the similarity score is calculated by the embedding similarity search. On the other hand, the structure-based retrieval is designed to fetch related neighbors from the graph with a novel hash-based Jaccard similarity estimation. The semantic similarity score and structural similarity score are merged to select the final neighbors, which are put into the LLM together with the center node for problem-solving. The authors then conduct experiments on three real-world datasets to demonstrate the effectiveness of their proposed method.

**Strengths:**

- This paper is very well-written and easy to follow.
- The proposed method of conducting LLM-based learning on graphs without a GNN component is novel and makes sense.
- The proposed hash-based structural similarity calculation is novel to me.

**Weaknesses:**

- Some model designs are not well-illustrated. For example, how the sampled neighbors and center text is finally fed into the LLM? What kind of instruction or prompt are you using? Do you train the model or just use do direct prompting?

- Some experiments on larger datasets or other tasks other than node classification can be helpful. The experiments are mainly focused on 10k-size graphs. Can the method be scaled to a large graph with millions of nodes? Node classification might not be enough to demonstrate the strength of the proposed method. It would be interesting to try on some more advanced LLM-based graph reasoning benchmarks [1].

- How many neighbors are finally selected? Is the model performance sensitive to the number of selected neighbors? Is there any scalability issue?

- Typos: (1) “where |S| is the size of the text-attributed nodes” should it be |V|? (2) The equation in line 224 needs one further “=” to be complete.



[1] Jin, B. Graph chain-of-thought: Augmenting large language models by reasoning on graphs. ACL 2024.

**Questions:**

See the weakness section.

---

> ### Author Response · Authors · 2024-11-23
> **Response to reviewer L5Yj**
>
> Dear Area Chair and Reviewers:
>
> Thank you for your commments and suggestions about our paper. Next are our detailed responses and explanations about each comments:
>
> **[W1]Model design**
>
> Thank you for your valuable suggestions. In Section 2.3, we mentioned "aggregated learning," but it needs to be clarified further. To address how the center text and sampled texts are fed into the LLM, we directly concatenate the center node text with the selected texts to form the unified input. For each node, we rely on the equation in line 269 to select the top K nodes based on semantic and structural similarity. Regarding the prompt design, we simply give the language model nomic the concatenated texts and the large language model llama3 that the task is node classification and specify the descriptions of label classes. Besides, we train the model, evaluating its performance after several epochs.
>
> **[W2]Datasets and different tasks**
>
> Thank you for your insightful feedback. We acknowledge the importance of testing our method on larger datasets. In response to your suggestion, we conducted experiments with a larger dataset, that is Arxiv dataset, and the results are as follows:
>
> | Models    | Datasets-Arxiv |
> |--------   | --------       |
> |GLEM       | 74.69%         |
> |LLAGA      | 75.85%         |
> |GraphFormer| 71.66%         |
> |InstructGLM| 75.70%         |
> |Our(nomic) | 76.18%         |
> |Our(llama3)| 76.53%         |
>
>
> Our model demonstrated the capability to handle graphs with a higher number of nodes effectively. Regarding other tasks, such as graph reasoning, we recognize that they differ significantly from classification tasks. Graph reasoning places a stronger emphasis on understanding and leveraging the relationships and requires a deeper analysis of how these relationships interact to infer new insights or answer complex queries. Your suggestion has inspired us to consider how our methodology could adapt to enhance graph reasoning capabilities.
>
>  **[W3]Selected nodes and scalarbility**
>
> We are appreciated of your questions. We rank all left nodes, excluding the anchor node, based on their scores of semantic and structural similarities, as described by the equation in line 296. The texts of these nodes are then concatenated according to their rank scores. The number of nodes to concatenate is determined by the maximum sequence length allowed by the language models. The number of nodes selected for concatenation can vary based on both the maximum sequence length the model accepts and the lengths of the text associated with datasets. This variation leads to different numbers of nodes being chosen for concatenation, affecting performances. Scalability remains a challenge primarily due to the LM's restrictions on text length. As the number of nodes increases, we face difficulties in incorporating all relevant node information within the language model's constraints.
>
> **[W4]Equation questions**
>
> Thank you for your valuable suggestions! You are correct that using |V| to represent the number of nodes is indeed more appropriate. We will make sure to revise that in our updates. Regarding the equation in line 224, I understand the confusion. It is meant to express the specific calculation of the function rather than a judgment or comparison, which is why a single “=” is used instead of “==”.

---

> > ### Comment · Reviewer_L5Yj · 2024-11-24
> >
> > Thank you for the reply. My questions are solved. I would encourage the author to add the new results on larger graphs and other tasks in the revised paper. I will keep my score.

---

### Comment · Area_Chair_ZdDo · 2024-11-25
**Please reply to the authors' response.**

Dear reviewers,

The ICLR author discussion phase is ending soon. Could you please review the authors' responses and take the necessary actions? Feel free to ask additional questions during the discussion. If the authors address your concerns, kindly acknowledge their response and update your assessment as appropriate.


Best,
AC

---

### Author Response · Authors · 2024-11-29
**Summary of our response**

Dear Area Chairs,

We appreciate all the reviewers and area chairs for their assessment and suggestions. We understand the effort required by area chairs to manage numerous papers; therefore, we've created an overview that summarizes our work, addresses reviewers' concerns, and provides additional clarifications to facilitate the evaluation process.


**Background and Contribution.**

Text-attribuetd graphs are frequently encountered in real life scenarios. Traditional pipleline employs NLP technique to embed text features and apply GNNs to propagate on the graph. We noted that current method adopts a two-staged framework, thus leading to inefficiency and not end-to-end disadvantage. Therefore, we aim to address TAGs only using language models. The main contributions are summarized as follows:

* We are the pioneer to transform TAG problems into a long context model-based problem.
* We proposed a novel ranking scheme assessing the relatedness of each node from both semantic and graph perspective.
* Our model introduces a hash-based mechanism to efficiently rank the structural relatedness based on common neighbors.
* With the help of useful content retrieved, the performance of language model only can achieve better performance than existing methods.


**Feedback from the reviewers.**

We appreciate the reviews recognizing the strengths of our papers, and here are the main points:

* The proposed method of conducting LLM-based learning on graphs without a GNN component is novel and makes sense.
* The paper is overall well-written and easy to follow. Our framework is well-motivated and novel.
* Using hash-based functions to simulate nodes intersections is effective and inspiring.

In response to reviewers’ comments, here are key highlights of how we address these requests.

* We want to clarify that our paper does not focus on RAG. Rather than in-context learning, we train the model specifically for TAG tasks. Our findings demonstrate that solely using LLMs does not yield optimal performance. However, we draw inspiration from RAG's concept of retrieving valuable content for each anchor node to enhance training. We will address this point and incorporate relevant related work and discussions in our paper.
* We provide a detailed explanation of the implementation and generalizability. Each component is easy to achieve, and the backbone is applicable to other variants. Also, we have shown our framework is not sensitive to hyperparameters and time efficient in previous replies.
* The new ranking mechanism requires only a few minutes and significantly reduces overall time, as it eliminates the need for GNNs.
* All typos and minor issues have been corrected.

We hope this overview helps our AC navigate the key information on this page. Additionally, we look forward to sharing our effective design and new insights into TAG learning with you and our fellow scholars in this important field.

Yours sincerely,

Authors

---

### Meta-Review · Area_Chair_ZdDo · 2024-12-18

**Metareview:**

The paper proposes a method of using large language models (LLMs) for text-attributed graphs (TAGs) without relying on Graph Neural Networks (GNNs). The strengths of the paper include its effective retrieval mechanism and promising performance on three datasets.

However, the paper has several weaknesses, such as unclear model design, scalability concerns, and limited empirical demonstration. The authors provided detailed clarifications and additional experiments in response to these concerns, but the issues remain.

Despite the authors’ efforts to address the reviewers’ questions, the paper still lacks clarity in model design and scalability to larger graphs. The limited evaluation on small datasets and marginal improvement over existing methods further weaken the paper’s contributions. Furthermore, the approach seems similar to RAG-based methods but it does not discuss this in the paper, leaving the novelty unclear.

Therefore, the paper is not recommended to be accepted to this conference.

**Additional Comments On Reviewer Discussion:**

Despite the authors’ efforts to address the reviewers’ questions, the paper still lacks clarity in model design and scalability to larger graphs. The limited evaluation on small datasets and marginal improvement over existing methods further weaken the paper’s contributions. Furthermore, the approach seems similar to RAG-based methods but it does not discuss this in the paper, leaving the novelty unclear.

---

### Decision · Program_Chairs · 2025-01-22

Reject